Processing of Noni Liquor based on response surface methodology

Gong Shusen
Yang Fei
Wang Qingfen
Wu Tian wutianpotato@swfu.edu.cn
Yunnan Province Engineering Research Center for Functional Flower Resources and Industrialization, Southwest Landscape Architecture Engineering Research Center of State Forestry and Grassland Administration, Southwest Forestry University, Southwest Forestry University , Kunming , Yunnan , China
Sistla Srinivas
Electronic publication date: 2022 Aug 16
Publication date: 2022
Volume: 10
Electronic Location ID: e13817
Received 2022 Feb 28; Accepted 2022 Jul 8
Copyright: ©2022 Gong et al.
Copyright year: 2022
Copyright holder: Gong et al.
License: This is an open access article distributed under the terms of the Creative Commons Attribution License, which permits unrestricted use, distribution, reproduction and adaptation in any medium and for any purpose provided that it is properly attributed. For attribution, the original author(s), title, publication source (PeerJ) and either DOI or URL of the article must be cited.
License URL: https://creativecommons.org/licenses/by/4.0/

Keywords: Noni, Polysaccharides, Polyphenois, Flavonoids, Ethanol extraction

Funding: Extension Project of State Administration of Forestry and Grassland, China ([2019]27) Key Research and Development Program of Yunnan Province, China 2019IB011 Science and Technology Innovation Fund Project of Southwest Forestry University, China KY21030 This research was supported by grants from the Extension Project of State Administration of Forestry and Grassland, China ([2019] 27), which provided experimental materials; the Key Research and Development Program of Yunnan Province, China (2019IB011), which provided sufficient funds to purchase reagent supplies; and the Science and Technology Innovation Fund Project of Southwest Forestry University, China (KY21030), which provided the experimental site and part of the experimental equipment. The funders had no role in study design, data collection and analysis, decision to publish, or preparation of the manuscript.

==============================
Noni (Morinda citrifolia L.) is widely used as a health food and medicine because it is rich in polysaccharides, polyphenols, and flavonoids; it is precisely because noni is rich in these substances that people usually like to soak it in liquor to drink. This study sought to maximize the main active constituents (polysaccharides, polyphenols, and flavonoid s) dissolved in liquor and spirits soaked with noni fruit, using ethanol as the solvent to optimize the extraction conditions using response surface methodology. The highest polysaccharide yield of 16.35% was found at 60 °C for 3.5 h, a liquid-solid ratio of 52 mL/g, and an ethanol concentration of 25%. The optimal extraction conditions for polyphenols were 75 °C for 1.5 h, a liquid-solid ratio of 90 mL/g, and an ethanol concentration of 50%, resulting in a 10.37% yield. The optimum extraction conditions for flavonoids were 75 °C for 1 h, a liquid-solid ratio of 70 mL/g, and an ethanol concentration of 100%, with 1.35% yield. Many active ingredients, including polysaccharides, polyphenols, and flavonoids, were obtained via alcohol extraction of noni fruit, suggesting that liquor and spirits soaked with noni fruit are novel and promising types. This study provides a technical reference for the production of high-quality noni liquor. It is recommended to use the best conditions for the extract of polysaccharides, polyphenols, and flavonoids, and blending them to obtain the desired degree of alcohol.

Introduction

Noni (Morinda citrifolia L.) is a perennial green tree species belonging to Rubiaceae. It is distributed in some tropical and subtropical coastal regions of South Pacific islands, Hawaii, Southeast Asia, West India, the Caribbean, and South America (Wu, Li & Lan, 2019). As a multifunctional plant, it is widely used as herbal medicine, cosmetic, nutraceutical food, animal feed, dyestuff, and lignum. Each part of noni such as flower, fruit, seed, bark, stem, leave, and root has a unique efficacy and value (Arunachalam, 2018). Notably, the noni fruit is an abundant source of diverse and valuable phytochemicals. The polysaccharides, polyphenols, and flavonoids fractions in noni fruit are of interest due to the superior medicinal value. The polysaccharides of noni fruit exhibit significant antitumor, anti-inflammatory, and antinociceptive activities (Jin et al., 2019). Polyphenols are the most active and standard pharmacological component of noni fruit with antioxidant, free radical-scavenging, anti-inflammatory, and antibacterial properties (Krishnaiah et al., 2015; Thomaz et al., 2018). In addition, the hepatoprotective (Nayak et al., 2011), immunomodulatory (Lohani, 2010), and antioxidant properties (Dussossoy et al., 2011) of noni fruit are attributed to its flavonoid content. The active ingredients extracted from noni are utilized in different fields such as medicine, toiletry, nutritional and health products.

Liquor and spirits can be used to effectively deliver active ingredients. Responsible drinking practices are another approach to maintain good health. Liquor and spirits are used universally as solvents to soak active ingredients derived from plants, animals, and insects dating back to ancient China. They are regarded as traditional preparations including medicinal liquors or tonic wines for healing and prophylactic applications (Nikolova et al., 2018), similar to tinctures used in western medicine. The medicinal liquors, such as notoginseng (Panax notoginseng) (Liao et al., 2017) and ginseng (Panax ginseng) (Li et al., 2019), occupy an important position in Chinese medicine and culture, and play an integral role in daily medical care in China or the Asia. For instance, individuals living in peat swamps in Narathivas province, Thailand, still use their unique tropical plants soaked in liquor for drinking or smearing on the diseased areas as febrifuges, antidiarrheas, tonics, and to treat skin injury (Kitirattrakarn & Anantachoke, 2003). Medicinal liquors derived from ginseng, Schisandrae fructus, Angelica sinensis, and Chrysanthemum indicum are popular in Korea (Kim et al., 2014). Additionally, in Poland, the bison grass was soaked in vodka used for flavoring alcohols, which represents the origin of Zubrowka Bison Brand Vodka (Baczek et al., 2014).

Medicinal liquor is very popular in China due to its simple preparation, low cost, and curative effect. With the introduction of noni in China, the knowledge and awareness of its health function have led to preparation of medicinal liquor with dried noni slices. However, improper preparation methods can diminish the efficacy of noni, which can prevent large-scale, standardized production. Ramamoorthy & Bono (2007) generated high content of total phenols and flavonoids by comparing different methods of high-pressure and ultrasonic extraction. Sasmito et al. (2015) evaluated the fractionation of noni polysaccharides using the precipitation method, and Jin et al. (2019) extracted polysaccharides of noni fruit with hot water. Nonetheless, few studies have investigated the technological conditions of noni fruit soaked in alcohol. Therefore, the appropriate conditions for extracting polysaccharides, polyphenols, and flavonoids from noni fruit should be investigated.

Bai et al. (2011a) and Bai et al. (2011b) used the response surface method (RSM) to optimize supercritical CO2 extraction of noni seed oil. In addition, they optimized and developed a simple and rapid blitzkrieg extraction procedure using RSM to increase the yield of hexane extraction from noni seed (Bai et al., 2011a; Bai et al., 2011b). Wang et al. (2021) optimized the extraction parameters of noni fruit tea polysaccharides and polyphenols using RSM. In this study, we investigated the temperature, extraction time, the concentration of ethanol, and the liquid–solid ratio using RSM based on single-factor experiments to determine the optimal conditions for polysaccharides, polyphenols, and flavonoids derived from ethanol soaked with a dry piece of noni fruit. These results facilitate the screening and processing of noni fruit liquor containing the most active ingredients, and promote the development of noni and medicinal liquor industry.

Materials & Methods

Chemicals and reagents

The chemicals used in this study were purchased from Sigma-Aldrich, including Folin-Ciocalteu reagent, tea polyphenols, and rutin standard. Ethanol, sulfuric acid, phenol, sodium nitrite, glucose, sodium carbonate, aluminum nitrate, sodium hydroxide, aluminum chloride, and sodium acetate were analytical grades.

Plant material

The ripe noni fruit was harvested from Yuanjiang, Yunnan Province, China, on August 17, 2019. The fruit was cut into slices of about eight mm and dried in a 60 °C oven until a constant weight was achieved. The dry noni slices were pulverized with an automatic pulverizer and sieved through an 80-mesh. The noni powders were stored in the dark at room temperature with silica-gel desiccant until used.

Extraction procedures

A single-factor design was used to identify the primary range of extraction process variables. The noni powders (0.5 g) were extracted in an ethanol solvent at different liquid–solid ratios (10 mL/g, 20 mL/g, 30 mL/g, 40 mL/g, 50 mL/g, 60 mL/g, 70 mL/g, 80 mL/g, and 90 mL/g) and concentrations (25%, 50%, 60%, 75%, and 100%) for varying times (0.5 h, 1 h, 1.5 h, 2 h, 2.5 h, and 3 h) at specific temperatures (30 °C, 40 °C, 50 °C, 60 °C, and 70 °C), and then sonicated (45 kHz) for 12 min. The crude extracts were separated from the residue via centrifugation (4,000 rpm, 10 min) and then filtered through Whatman No. 1 filter paper. The extraction of the residue was repeated under the same conditions.

Experimental design

The extraction optimization of noni polysaccharides, polyphenols, and flavonoids was carried out under three operational parameters of Box–Behnken design (BBD): liquid–solid ratio, extraction time, and extraction temperature. The response factors coded as X1, X2, and X3, were evaluated at three levels coded as −1, 0, and 1. The range of parameters was confirmed via single-factor experiments, and the ethanol concentration was fixed at 25%, 50%, and 100% for polysaccharides, polyphenols, and flavonoids, respectively, resulting in the highest yield. Table 1 shows the ranges and medians of the three operational parameters based on the above experimental results.

Table 1 The levels of parameters of BBD for extraction experimental.

Parameters	Codes	Levels	
			−1	0	1	
Extraction of polysaccharide	Liquid–solid ratio (mL/g)	X1	40	50	60	
Temperature (°C)	X2	40	50	60	
Time (h)	X3	2.5	3.0	3.5	
Extraction of polyphenol	Liquid–solid ratio (mL/g)	X1	70	80	90	
Temperature (°C)	X2	65	70	75	
Time (h)	X3	1.5	2.0	2.5	
Extraction of flavonoid	Liquid–solid ratio (mL/g)	X1	70	80	90	
Temperature (°C)	X2	65	70	75	
Time (h)	X3	1.0	1.5	2.0	

Determination of extraction yield

The yield of polysaccharides, polyphenols, and flavonoids (%) was calculated using the following formula: (1) Extractionyield%=C×N×VW×100%.

In the formula, C is the concentration of polysaccharides, polyphenols, and flavonoids calculated using the calibrated regression equation (µg/mL); N denotes the dilution ratio; V represents the total volume of crude extract solution (mL), and W is the weight of noni powders (g).

Determination of total polysaccharide content

The total polysaccharide content of noni extract was determined via slight modification of phenol–sulfuric acid method (Han et al., 2016), using glucose as the standard. Each 0.1 mL sample was mixed with 0.9 mL of distilled water and one mL of 5% phenol after rapid mixing with 3.5 mL of concentrated sulfuric acid. The mixture was heated at 100 °C for 15 min, and then cooled down to room temperature. Next, the UV absorption at 490 nm was monitored, and the straight line was represented by the equation Y = 0.012X+0.0213, R2 = 0.9953.

Determination of total polyphenol content

The total polyphenol content of noni extract was determined using a slightly modified Folin-Ciocalteu method (Hannachi et al., 2019), taking tea polyphenols as the standard. Each 0.5 mL sample was mixed with 0.5 mL of distilled water and one mL of Folin-Ciocalteu reagent, and allowed to stand for 5 min. Next, two mL of 10% sodium carbonate solution was added and left in the dark at room temperature for about 60 min. Finally, the UV absorption at 760 nm was monitored, and the straight line was represented by the equation Y = 0.0044X+0.09, R2 = 0.9991.

Determination of total flavonoid content

The total flavonoid content was determined using a slightly modified colorimetric method (Zheng et al., 2019), and rutin as the standard. Each five mL sample was mixed with 0.3 mL of 5% sodium nitrite solution and maintained for 6 min, and subsequently mixed with 0.3 mL of 10% aluminum chloride solution followed by the addition of two mL of 4% sodium hydroxide solution after 6 min. The solution was mixed well and allowed to sit for 10 min. The UV absorption at 504 nm was monitored, and the straight line was represented by the equation Y = 0.0041X+0.0629, R2 = 0.9979.

Statistical analysis

Statistical analysis was conducted using IBM SPSS Statistics (Version 22) to determine the effect of individual factors on the yield of polysaccharides, polyphenols, and flavonoids (single-factor experiment). Design-Expert software (Version 8.0.6) was used to construct Box–Behnken design and assess the effect of linear and quadratic interactions of the operational parameters on the extraction of polysaccharides, polyphenols, and flavonoids. All tests were performed in triplicate.

Results and Discussion

Results of one-factor experiment

In this single-factor experiment, the polysaccharide extraction yield was decreased by the increase in ethanol concentration (Fig. 1A). This is consistent with the study of Samavati that is as the concentration of alcohol increases, the viscosity of the alcohol solution increases, and the extraction rate of polysaccharides decreases. (Samavati, 2013). The extraction time was 3 h when the ethanol concentration was 25%, and the extraction temperature was 50 °C. The polysaccharide extraction rate was the highest at a liquid–solid ratio of 40 mL/g (Figs. 1B, 1C and 1D).

Figure 1 (A–D) The results of a one-way experiment.

High levels of ethanol concentration decreased the polyphenol yield. A similar phenomenon was reported during the extraction of polyphenols from olive (Martínez-Patiño et al., 2019) and sunflower seed cake (Zardo et al., 2019) under the optimal conditions of 50% (Fig. 1A). The optimal extraction temperature, extraction time, and liquid–solid ratios were 70 °C, 2 h, and 80 mL/g, respectively (Figs. 1B, 1C and 1D).

The extraction yield of flavonoids was significantly elevated at the highest ethanol concentration (Fig. 1A). Compared with a study by Yusof et al. (2019) who reported that the flavonoid yield increased with the ethanol concentration and reduced after reaching a concentration of 80%, the disparity could be explained by the difference in material. Likewise, when the extraction temperature was at its highest, the extraction rate was also at its peak (Fig. 1B). The optimal extraction time and liquid–solid ratio were 1.5 h and 80 mL/g, respectively (Figs. 1C and 1D).

These data were employed as fixed parameters in the optimization experiment.

Optimization of polysaccharide conditions

The extraction yield of polysaccharides was considered as a dependent variable in the Box–Behnken design. Under the experimental conditions of 15 runs, the results from polysaccharides varied in the range of 14.42% to 16.25% (Table 2). The regression equation of the predicted response Y for polysaccharides yield was obtained as follows: (2) Y=15.50+0.041X1+0.15X2+0.0025X3+0.24X1X2−0.048X1X3+0.78X2X3−0.75X12−0.022X22−0.060X32.

Table 2 Design and results of RSM experiment for extraction yield of polysaccharide, polyphenol and flavonoid.

No.	Coded parameter levels	Observed	Predicted	
	X1 (mL/g)	X2 (°C)	X3 (h)	Yield of polysaccharide (%)	Yield of polyphenol (%)	Yield of flavonoid (%)	Yield of polysaccharide (%)	Yield of polyphenol (%)	Yield of flavonoid (%)	
1	−1	0	−1	14.61	8.3	1.09	14.6	8.34	1.12	
2	0	0	0	15.6	8.49	1.04	15.5	8.55	1.03	
3	1	1	0	15.27	9.23	0.99	15.16	9.41	1.01	
4	0	0	0	15.24	8.61	1.1	15.5	8.55	1.03	
5	−1	−1	0	14.67	7.95	0.89	14.78	7.77	0.9	
6	0	0	0	15.66	8.54	0.99	15.5	8.55	1.03	
7	0	1	−1	14.81	9.57	1.23	14.78	9.43	1.21	
8	1	0	−1	14.64	10.09	0.98	14.78	10.05	0.99	
9	−1	0	1	14.84	8.39	0.98	14.7	8.43	0.97	
10	1	0	1	14.68	9.08	1.09	14.69	9.04	1.06	
11	0	−1	−1	16.15	8.44	0.92	16.05	8.58	0.89	
12	−1	1	0	14.56	8.21	1.18	14.6	8.31	1.18	
13	0	1	1	16.25	8.75	0.96	16.35	8.61	0.98	
14	0	−1	1	14.46	8.35	1.03	14.49	8.49	1.04	
15	1	−1	0	14.42	9.09	1	14.38	8.99	1.03	

The significance and fit of the model were interpreted via analysis of variance (Table 3). Statistical significance (p < 0.05) suggested the construction of the final predictive model (Mojerlou & Elhamirad, 2018). The RSM analysis indicated that the model was significant (p < 0.05) for polysaccharides, although the lack-of-fit of the model was not significant (p > 0.05), which suggested that the experiment and the response surface quadratic model fit satisfactorily, and the extraction yield of polysaccharides could be analyzed and predicted using this model within the design range. The coefficient of determination was 0.9628, which was closer to 1, meaning a better association between the observed and predicted values (Pan et al., 2012). Further, the model performance was also evaluated using the coefficient of variation, which was 1.3% (C.V. <5%), suggesting that the experimental values were highly accurate and reliable. The terms X2X3 and X12 were highly significant (p < 0.001), indicating that the interaction of temperature with time played a key role in polysaccharide extraction and the quadratic term of liquid–solid ratio dominated the exponential change in the extraction yield.

Table 3 Variance analysis for fitting quadratic model of extraction yield of polysaccharide, polyphenol and flavonoid.

Source	Sum of square	Degree of freedom	Mean square	F-value	P-value	
Extraction of polysaccharide						
Model	4.95	9	0.55	14.39	0.00045**	
X1	0.014	1	0.14	0.36	0.5766	
X2	0.018	1	0.18	4.63	0.0840	
X3	5.000E−005	1	5.000E−005	1.309E−003	0.9725	
X1X2	0.23	1	0.23	6.03	0.0575	
X1X3	9.025E−003	1	9.025E−003	0.24	0.6475	
X2X3	2.45	1	2.45	64.11	0.0005**	
X12	2.06	1	2.06	54.00	0.0007**	
X22	1.869E−003	1	1.869E−003	0.049	0.8337	
X32	0.013	1	0.013	0.35	0.5809	
Residual	0.19	5	0.038			
Lack of fit	0.088	3	0.029	0.57	0.6883	
Pure error	0.10	2	0.052			
Correlation Total	5.14	14				
R2	0.9628	Adj R2	0.8959			
C.V.%	1.30	Pred R2	0.6813			
PRESS	1.64	Adeq Precision	12.336			
Extraction of polyphenol						
Model	4.39	9	0.49	13.28	0.0052*	
X1	2.69	1	2.69	74.88	0.0003**	
X2	0.47	1	0.47	12.96	0.0156*	
X3	0.42	1	0.42	11.65	0.0190*	
X1X2	3.600E−003	1	3.600E−003	0.10	0.7644	
X1X3	0.30	1	0.30	8.42	0.0337*	
X2X3	0.13	1	0.13	3.71	0.1122	
X12	0.063	1	0.063	1.75	0.2434	
X22	0.012	1	0.012	0.33	0.5879	
X32	0.31	1	0.31	8.52	0.0331*	
Residual	0.18	5	0.036			
Lack of fit	0.17	3	0.057	15.82	0.0600*	
Pure error	7.267E−003	2	3.633E−003			
Correlation Total	4.57	14				
R2	0.9607	Adj R2	0.8900			
C.V.%	2.17	Pred R2	0.3931			
PRESS	2.78	Adeq Precision	14.730			
Extraction of flavonoid						
Model	0.11	9	0.012	5.96	0.0318*	
X1	8.000E−004	1	8.000E−004	0.39	0.5599	
X2	0.034	1	0.034	16.46	0.0098*	
X3	3.200E−003	1	3.200E−003	1.56	0.2672	
X1X2	0.022	1	0.022	10.96	0.0212*	
X1X3	0.012	1	0.012	5.89	0.0596	
X2X3	0.036	1	0.036	17.58	0.0085*	
X12	7.410E−004	1	7.410E−004	0.36	0.5742	
X22	7.410E−004	1	7.410E−004	0.36	0.5742	
X32	1.256E−004	1	1.256E−004	0.061	0.8145	
Residual	0.010	5	2.053E−003			
Lack of fit	4.200E−003	3	1.400E−003	0.46	0.7383	
Pure error	6.067E−003	2	3.033E−003			
Correlation Total	0.12	14				
R2	0.9147	Adj R2	0.7612			
C.V.%	4.39	Pred R2	0.3283			
PRESS	0.081	Adeq Precision	8.919			
Notes.

* Means significant at P < 0.05.

** Means significant at P < 0.001.

The 3D response plot showed that the polysaccharides yield increased rapidly with the increase in the liquid–solid ratio of 40–50 mL/g, and the polysaccharide yield decreased above the liquid–solid level of 50 mL/g (Figs. 2A and 2B). In addition, the polysaccharide yield along with the temperature increased under 3 h at a liquid–solid ratio of 50-60 mL/g (Fig. 2A). Temperature positively affects the transfer efficiency of the water-soluble polysaccharides from cells (Qu et al., 2016). The liquid–solid ratio of 50 mL/g and the extraction time of 2.5 h contributed to the high polysaccharides yield of 16.15% when the temperature was reduced to 40 °C (Fig. 2C). Conversely, the liquid–solid ratio of 50 mL/g and the temperature of 60 °C contributed to the highest polysaccharide yield of 16.25% when the extraction time was increased to 3.5 h. This result suggested that the interaction between temperature and time significantly affected polysaccharide extraction. An increase in temperature and duration of extraction enhanced the polysaccharide yield as reported by Chaiwut et al. (2019).

Figure 2 The 3D response surface plot and contour plot of polysaccharide extraction yield.

Optimization of polyphenol conditions

The polyphenol levels changed from 8.21% to10.09% (Table 2). A second-order polynomial equation of the predicted response Y for polyphenol yield was as follows: (3) Y=8.55+0.58X1+0.24X2−0.23X3−0.030X1X2−0.27X1X3−0.18X2X3+0.13X12−0.057X22+0.29X32.

This model was significant (p < 0.05) for polyphenols, and the lack of fit of the model was not significant (p > 0.05) (Table 3). The coefficient of determination was 0.9607, and the coefficient of variation was 2.17% (C.V. <5%). In addition, the terms X1, X2, X3, X1X3, and X32 were significant, indicating that the role of the three-factor variables (liquid–solid ratio, temperature, and time) was vital in polyphenol extraction.

The effects of extraction temperature and liquid–solid ratio on extraction rate showed a similar trend (Fig. 3A). The extraction time showed a significant interaction with the liquid–solid ratio; in the response surface, the polyphenol extraction was enhanced by a higher liquid–solid percentage under time. An increase in liquid–solid ratio from 70 mL/g to 90 mL/g, or a decrease in extraction time from 2.5 h to 1.5 h boosted the polyphenol yield with a maximum recovery of 10.09% at 70 °C (Fig. 3B). Yusof et al. (2019) defined a region of low response variability as an optimal region, which was supposed to be within the liquid–solid ratio above 80 mL/g, a temperature higher than 70 °C and between 1.5 and 2 h. Azahar, Gani & Mokhtar (2017) reported a similar maximum yield of phenolics at the higher temperature of 75−80 °C.

Figure 3 The 3D response surface plot and contour plot of polyphenol extraction yield.

At the same time, the polyphenols extraction time was reduced to 80-100 min. Additional extraction time does not necessarily lead to higher extraction rate (Fig. 3C), which might be attributed to the change of adsorption surface and the diffusion of additional polyphenols into the solvent with high liquid-to-solid ratio (Hannachi et al., 2019). The high temperature might soften the cell wall, hydrolyze polyphenols, increase the diffusion coefficient and enhance the solubility of polyphenols. The short extraction time prevents the degradation of polyphenols and reduces the types of polyphenols extracted (Sai-Ut et al., 2015).

Optimization of flavonoid conditions

The extraction ratio of flavonoids was in the range of 0.89−1.09% (Table 2). The mathematical models were used to predict flavonoid yield using the following second-order polynomial equation: (4) Y=1.04−0.010X1+0.065X2−0.020X3−0.075X1X2+0.055X1X3−0.095X2X3−0.014X12−0.014X22+0.005833X32

In addition to the lack of fit (p > 0.05), a determination coefficient of 0.9147 and coefficient of variation (C.V. <5%) fully demonstrated the reliability of the model. The interaction between liquid–solid ratio with temperature (X1X2) and temperature with time (X2X3) had a significant impact on flavonoid extraction (p < 0.05). The interaction between liquid–solid ratio and time has little effect on the extraction rate (Fig. 4B). The temperature showed a significant effect on the extraction yield of flavonoids. Increase in temperature from 65 to 75 °C (for an extraction time of 1 h under liquid–solid ratio 80 mL/g) increased the flavonoid yield from 0.92% to 1.23%, or the increase in temperature from 65 to 75 °C under an extraction time of 1.5 h and a liquid–solid ratio of 70 mL/g, with the flavonoid yield increasing from 0.89% to 1.13% (Figs. 4A and 4C). With other conditions unchanged, the increase in temperature led to an increase in the yield. Mohd-Setapar et al. (2014) reported that the higher temperature had a positive effect on the diffusion of the solute, the viscosity and surface tension of the solution, the solubility of the target compound in the solvent, and the extraction efficiency. In addition, the extraction time and the liquid–solid ratio were not linearly correlated with the extraction yield of flavonoids. The 3D response plot indicated that a liquid–solid ratio below 80 mL/g, a temperature higher than 70 °C and extraction time ranging between 1 and 1.5 h were the optimal parameters.

Figure 4 The 3D response surface plot and contour plot of flavonoid extraction yield.

Experimental validation of optimal conditions

The optimal conditions for the extraction of polysaccharides, polyphenols, and flavonoids including the liquid–solid ratio, extraction temperature, and extraction time, were determined from the constructed model. To determine the validity of the quadratic model for predicting the yield of polysaccharides, polyphenols, and flavonoids, experimental verification was performed under optimal conditions, and the yields were confirmed. The predicted extraction yield of polysaccharides was 16.37% with 25% ethanol at 60 °C for 3.5 h, and the liquid–solid ratio was 51.88 mL/g. Considering the actual production conditions, the liquid–solid ratio was revised to 52 mL/g, resulting in an experimental yield of 16.35 ± 0.05%. The predicted extraction yield of polyphenols was 10.39% with 50% ethanol at 75 °C for 1.5 h, and the liquid–solid ratio was 90 mL/g. The experimental yield of polyphenols was 10.37 ± 0.05%. Further, the predicted extraction yield of flavonoids was 1.34% with 100% ethanol at 75 °C for 1 h, and the liquid–solid ratio was 70 mL/g. The experimental yield of flavonoids was 1.35 ± 0.02%. All the predicted results were consistent with the experimental results, suggesting model reliability for optimization of extraction.

Lack of experimentation

In the model constructed in this study (Table 3), the Adj R2 values were less than 0.9 for polysaccharides, polyphenols, and flavonoids. The Pred R2 values were also not in reasonable agreement with the Adj R2, in which the difference was greater than 0.2, although the R2 values were closer to 1. Anomalies in these values suggest model inadequacy (Koocheki et al., 2009; Che Sulaiman et al., 2017), which may be due to experimental error. The dependent variable in the quadratic regression model was the mean value of the experimental data, and there were no repeated groups. Using stepwise regression and after deleting non-significant terms, in case of polysaccharides, the value of R2 was 0.9582; the value of Adj R2 was 0.9269, and the value of Pred R2 was 0.8657. Likewise, in polyphenols and flavonoids, the values tended to be normal. The values of Adj R2 was also similar to the values of Pred R2 after deletion of insignificant terms, suggesting that the model was reasonable. However, the reduced regression quadratic model failed to display a few interactions. Therefore, we still preferred to use the original model for response surface analysis.

Conclusions

The overall results indicated the feasibility and success of RSM for optimal extraction of active ingredients in noni via RSM, which was accurate and reliable in identifying the optimal parameters for the extraction yield of polysaccharides, polyphenols, and flavonoids from noni fruit. Ethanol concentration, extraction temperature, time, and liquid–solid ratio were essential parameters influencing extraction when noni fruit soaked in liquor was used to obtain polysaccharides, polyphenols, and flavonoids. The use of noni powders was recommended for rapid preparation of noni fruit liquor. To obtain noni fruit liquor with high levels of polysaccharides, polyphenols, and flavonoids, we recommend the best extraction conditions according to the 3D response plot, followed by a combination of the active ingredients. Finally, the mixture was adjusted to the appropriate alcohol content (ethyl alcohol number of degrees). The alcohol content, polysaccharides, polyphenols, and flavonoids in the mixture is clearly visible. The optimized extraction processes reported in the study can be used to manufacture noni fruit liquor commercially.

Supplemental Information

Data S1 Raw data

Click here for additional data file.

Thanks for the support of the Foundation and the contributions of the authors.

Additional Information and Declarations

Competing Interests

Author Contributions

Data Availability

The authors declare there are no competing interests.

Shusen Gong conceived and designed the experiments, performed the experiments, analyzed the data, prepared figures and/or tables, and approved the final draft.

Fei Yang conceived and designed the experiments, performed the experiments, analyzed the data, prepared figures and/or tables, and approved the final draft.

Qingfen Wang performed the experiments, prepared figures and/or tables, and approved the final draft.

Tian Wu conceived and designed the experiments, authored or reviewed drafts of the article, and approved the final draft.

The following information was supplied regarding data availability:

The raw measurements are available in the Supplemental File.

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
