# Peer review of "Processing of Noni Liquor based on response surface methodology"

_PeerJ, doi:10.7717/peerj.13817_

## Round 0.1 · original submission · Major Revisions

The reviewer reports are available for your manuscript and I would like to bring this to your notice. There are major concerns expressed by the reviewers including English language editions are required before moving forward or reconsidering.

Reviewer 1 ·

Basic reporting

The study employs response surface methodology (RSM) to investigate the optimum extraction conditions of active constituents from noni, using ethanol as the extraction solvent, for the application of noni liquor. The extraction parameters included in the study are temperature and time of extraction, liquid to solid ratio, and ethanol concentration, to achieve the maximum yield of polysaccharides, polyphenols, and flavonoids from noni powder.

The manuscript was structured according to the PeerJ guidelines.
The data was sufficient, although it could be presented better.
For example:
- The first row in table 1 should be given a bottom border, and “Codeds” -> should be “Codes”
- First row in Table 2 : “Observer” -> should be “Observed”
- Incorrect caption for Table 3 -> it was written “Table 1”
- The result of the single-factor experiment was not displayed in the form of graph. It would help readers to understand better if the author provided the graphs showing the trend for the single-factor experiments.

The literature references provide sufficient context and background. However, since this manuscript reported the optimization of noni extraction using RSM, I think it would be better if the authors also include the previous study related to the optimization of noni extraction using RSM.

The authors must proofread their manuscript carefully and thoroughly. English of the article requires some polishing, and needs to be reworked for better understanding for the readers.
For example:
Line 178-179 : “When the terms are significant (p < 0.05), the final predictive model can be regarded as to construct” -> it seems like an unfinished sentence.
Line 201: enhance -> enhancement
Line 190-201 : difficult to understand
212-226 : difficult to understand
233-244 : difficult to understand
255-259 : difficult to understand, incorrect use of conjunction

Experimental design

The present study is generally similar to the previous study by the same research group (Wang, Q., Yang, F., Jia, D. and Wu, T., 2021. Polysaccharides and polyphenol in dried Morinda citrifolia fruit tea after different processing conditions: Optimization analysis using response surface methodology. PeerJ, 9, p.e11507), which reported the optimization of extraction conditions of polysaccharides and polyphenols from noni, using water as the extraction solvent, for the application of noni tea.

The sample preparation and analysis step, as well as the extraction parameters, are the same as the previous one.

The difference from the previous publication is the change of extraction solvent from water to ethanol, the addition of flavonoids content as the response, and the addition of a single-factor approach to identify the primary range of extraction process variables.

Validity of the findings

In the result and discussion section, the authors mentioned that the constructed model was significant and the lack-of-fit was insignificant. However, this is not conclusive evidence that the models accurately represent the data in the experimental region.
The authors also mentioned the coefficient of determination (R2) but did not discuss the Adj R2 and Pred R2 values (listed in Table 3).

A high R2 value indicates that the variation could be accounted for by the data satisfactorily fitting the model. However, a high value of R2 does not always imply that the regression model is a good one. Adding a variable to the model will always increase R2, regardless of whether the additional variable is statistically significant or not. Thus, it is better to use an Adj R2 to evaluate the model adequacy.
(Samavati, V., 2013. Polysaccharide extraction from Abelmoschus esculentus: Optimization by response surface methodology. Carbohydrate polymers, 95(1), pp.588-597.)

Adj R2 indicates the descriptive power of the regression models while including the diverse numbers of variables. Therefore, considering the Adj R2 value is important to evaluate the adequacy of the model because the value only increases if the variables enhance the model beyond what would normally be obtained by probability. Adj R2 values above 0.9 may be used to indicate the adequacy of the model. Furthermore, a difference of less than 0.2 between Adj R2 and Pred R2 demonstrates the effectiveness of the model. (Sulaiman, 2017)
(Koocheki, A., Taherian, A.R., Razavi, S.M. and Bostan, A., 2009. Response surface methodology for optimization of extraction yield, viscosity, hue and emulsion stability of mucilage extracted from Lepidium perfoliatum seeds. Food Hydrocolloids, 23(8), pp.2369-2379.
Che Sulaiman, I.S., Basri, M., Fard Masoumi, H.R., Chee, W.J., Ashari, S.E. and Ismail, M., 2017. Effects of temperature, time, and solvent ratio on the extraction of phenolic compounds and the anti-radical activity of Clinacanthus nutans Lindau leaves by response surface methodology. Chemistry Central Journal, 11(1), pp.1-11.)

Also, Adj R2 is usually in an approximate value of R2 with differences that are no more than 0.1 to show the adequacy of the models obtained. If Adj R2 is significantly lower than R2, then it is possible that one or more explanatory variables are missing from the model.
(Thoo, Y.Y., Ho, S.K., Abas, F., Lai, O.M., Ho, C.W. and Tan, C.P., 2013. Optimal binary solvent extraction system for phenolic antioxidants from mengkudu (Morinda citrifolia) fruit. Molecules, 18(6), pp.7004-7022.)

According to the Table 3 in the present study, The Adj R2 value is less than 0.9 for polysaccharide and polyphenol, even lower (less than 0.8) for flavonoid.
The Pred R² values are also not in reasonable agreement with the Adj R², in which the difference is more than 0.2.

Therefore, the authors probably should consider the removal of non-significant terms from the full regression quadratic model using a backward elimination process. Because, in some cases, it might influence the significance of the experimental variables. It might also affect the regression coefficients, to a lesser extent.
And finally, depend on the result of the regression coefficient, it might be worth to try to use the reduced regression quadratic models with enhanced experimental variables to interpret the experimental data.
And if not, the author should still mention justification of using the developed model, despite the poor value of the Adj R2 and Pred R2.

Reviewer 2 ·

Basic reporting

1. The English language should be improved to ensure that an international audience can clearly understand your text. Some examples where the language could be improved include lines 44, 51, 84, 127, 134,138,143,201 – the current phrasing makes comprehension difficult. I suggest you have a colleague who is proficient in English and familiar with the subject matter review your manuscript, or contact a professional editing service.
2. No reference is made in the text to Figures 2a, 2c, or 3b.

Experimental design

Some of the phrase’s need be re written to express the right sense of what the authors intend to say. The statement “This might be attributed to the increase in the viscosity of the solution as the alcohol concentration increased, and the lower the viscosity could afford a higher extraction efficiency of the mucilage for polysaccharides” should be modified and written clearly to indicate what the authors want to say.
Similarly, statement “As a result, most of the polyphenols in noni extract were possibly polar, polar chemical structure contained one or more hydrophilic hydroxyl groups” and statement “In addition, the polysaccharides yield heightened along with the temperature was increased under three h of time, 50-60 mL/g of liquid-solid ratio” should also be written again.

Validity of the findings

No comments

---

## Round 0.2 · Minor Revisions

Please consider these changes.

Reviewer 1 ·

Basic reporting

The authors have given adequate responses to the raised concerns. Substantially, this study shows some merit and could be published.
However, I think some of the English still need to be further polished to meet the standard of PeerJ.
Here are some examples, just to name a few :

Line 17-18 : Noni (Morinda citrifolia L.) is widely used as a health food and medicine because it is rich in polysaccharides, polyphenols, and flavonoids; therefore, noni fruit is soaked with liquor. -> weird sentence.

Line 23 : The optimal extraction conditions for polyphenols 75°C for 1.5 h, a liquid-solid ratio of 90 mL/g, and an ethanol concentration of 50% -> missing “were / are”?

Line 30-31 : It is recommended to use the best conditions for the extract of polysaccharides, polyphenols, and flavonoids, and blending the to obtain the desired degree of alcohol -> blending the what?

Line 86 : ….. and flavonoids derived from ethanol soaked with a dry slice of noni fruit. -> incorrect word choice

Line 186 : Statistical significance (p < 0.05), the construction of the final predictive model. -> incomplete sentence.

Line 194 : …., revealing suggesting that the experimental values were highly accurate and reliable. -> redundant (... revealing suggesting ....)

Experimental design

There is no concern regarding the experimental design.

Validity of the findings

The major concerns regarding the validity of the findings have been addressed.

---

## Round 0.3 · accepted · Accept

All the concerns raised by the reviewers were addressed and I am happy to inform you that the paper is accepted for publication.

Reviewer 1 ·

Basic reporting

The concern regarding the English of the manuscript has been addressed.

Experimental design

There is no concern with the experimental design

Validity of the findings

No concern has been found regarding the validity of the findings